# Long-Term Outcomes of Canaliculotomy with Silicone Tube Intubation in the Management of Canaliculitis

**DOI:** 10.3390/jcm11226830

**Published:** 2022-11-18

**Authors:** Jens Julian Storp, Julian Alexander Zimmermann, Eliane Luisa Esser, Martin Dominik Leclaire, Nicole Eter, Maged Alnawaiseh, Ralph-Laurent Merté, Nataša Mihailovic

**Affiliations:** 1Department of Ophthalmology, University of Muenster Medical Center, 48149 Muenster, Germany; 2Department of Ophthalmology, Klinikum Fulda gAG, University of Marburg, Campus Fulda, 36043 Fulda, Germany

**Keywords:** canaliculitis, canaliculus, nasolacrimal duct obstruction, dacryoliths, actinomycetes, lacrimal duct concretions

## Abstract

As a rare and often misdiagnosed disease of the lacrimal apparatus, only limited data exist on long-term outcomes of surgical methods for the treatment of primary canaliculitis. The aim of this study was to evaluate canaliculotomy with silicone tube intubation (STI) as a surgical procedure for canaliculitis in a long-term follow-up setting. A total of 25 eyes of 25 patients with canaliculitis treated with canaliculotomy and STI at the University of Muenster Medical Center, Germany, from 2015 to 2021 were included in this study. Data analysis involved clinical symptoms, complications, duration of STI and rate of recurrence. Mean patient age was 63.7 ± 17.2 years. After a follow-up time of 3.7 ± 1.5 years, 88% of cases showed no recurrence of inflammation. The mean duration of STI was 5.8 ± 3.4 months. Complications such as post-operative hemorrhage, spread of infection, obstruction of the canaliculus or migration of the STI were not observed in any of the patients. However, tube dislocation occurred in four cases, a pyogenic granuloma in two cases and a fistula formation in one case. The management of canaliculitis by canaliculotomy with STI showed very good postoperative outcomes and low complication rates in the long-term and can therefore be considered a safe and successful surgical approach.

## 1. Introduction

Canaliculitis is a rare inflammatory disease of the lacrimal apparatus that can affect both the upper and lower canaliculus and/or the lacrimal punctum [1]. It accounts for about 2–4% of all lacrimal diseases and presents a diagnostic challenge in clinical practice [1,2]. Accurate measures of the prevalence of primary canaliculitis in the general population are unavailable, owing to the frequency of misdiagnosis, relative rarity and general under-reporting of the disease [1].

Typical symptoms of canaliculitis include redness and swelling in the area of the medial canthus with or without tenderness, epiphora, refractory or recurrent conjunctivitis, dilation or inverting of the lacrimal punctum with or without secretion of mucopurulent discharge, and the expression of concretions from the lacrimal punctum [1,3,4,5] (Figure 1A–C). Being the result of an incidental infection, symptoms are usually unilateral, but they can also occur binocularly [2].

Due to its rarity and variety of symptoms, canaliculitis is often misdiagnosed as conjunctivitis, blepharitis, chalazion, dacryocystitis or other diseases of the eyelid and nasolacrimal duct that occur more frequently in clinical practice, and it is consequently treated inadequately or incorrectly [1,6,7,8].

The average patient age is approximately 59 years. With a ratio of 5:1, women are affected much more frequently than men [1].

Canaliculitis is usually caused by a pathogen-related infection but can also occur iatrogenically after intervention, for example after insertion of a punctum plug [1]. The most common pathogen of canaliculitis is an anaerobic filamentous gram-positive organism, Actinomyces israelii, whose sulfur granules typically present as yellowish dacryoliths. However, in addition to actinomycetes, a variety of other pathogens, including staphylococci, streptococci, mycobacteria and pseudomonads, but also viruses and fungi, can cause canaliculitis [1].

While conservative treatment options such as local or systemic antibiotic therapy or sac washouts are often associated with persistence and recurrence of the disease, surgical treatment with removal of any concretions has been proven to be the approach of choice for long-term therapeutic success [3,7,8]. Possible surgical procedures include canaliculotomy with and without sparing of the lacrimal punctum, with and without curettage and with and without STI, as well as more minimally invasive methods like punctoplasties with and without combined (lateral) canaliculotomy [4,5,7,9,10,11]. However, none of the procedures mentioned can currently be regarded as the gold standard, mainly because studies in the literature are inconsistent in terms of inclusion criteria, definition of failure/recurrence and follow-up time [4]. Due to the rarity of the disease, there is also little data on the superiority of the different surgical approaches regarding their long-term outcomes. A prospective study by Landmann-Vu et al. investigating 96 patients with canaliculitis was able to show a statistically significant higher success rate of one-snip punctoplasty with lateral canaliculotomy compared to punctum-sparing canalicular curettage using a chalazion curette. Meanwhile, Wang et al. demonstrated a higher success rate for canaliculotomy with STI compared to canaliculotomy without STI. The choice of surgical procedure therefore remains left to the surgeon’s preference at the current time [4,8,11].

The primary aim of the present study was to evaluate the long-term outcomes of canaliculotomy and STI in patients with canaliculitis at a tertiary eye care center specializing in lacrimal duct surgery in Germany.

## 2. Materials and Methods

A retrospective analysis of adult patients with canaliculitis who underwent canaliculotomy and STI at the Department of Ophthalmology of the University of Muenster Medical Center, Germany, between 1 January 2015 and 31 December 2021 was performed.

None of the patients included in this study had any previous surgical treatment. A total of 17 patients were previously treated with topical antibiotic eye drops, while 5 patients had received topical and systemic antibiotic medication before visiting our clinic. For the evaluation, the combination of the diagnosis “canaliculitis” and the International Statistical Classification of Diseases (ICD-10) codes H04.3 and H04.4 as well as the operation and procedure codes (OPS) 5-084.11 and 5-086.31 were compiled in the digital findings documentation system FIDUS (Arztservice Wente GmbH, Darmstadt, Germany).

The study was approved by the Ethics Committee of the Westphalian Wilhelms University Münster and complies with the principles of the Declaration of Helsinki.

Information on age at diagnosis, gender, clinical symptoms, therapeutic procedure, occurrence of complications, duration of tube intubation and recurrence in the long-term were taken from the electronic patient records. Data on recurrence and duration of tube intubation were collected by telephone interview in those cases where no data were available in the patient file or patients did not return to our clinic after inpatient discharge. Tube dislocation was defined either as an unintended complete removal of the tube or an advanced dislocation that made manual repositioning of the tube impossible. Surgical success was equated with the absence of recurrent symptoms at the time of the follow-up. Failure was defined as the recurrence of initial symptoms at any time during the follow-up-period.

The data were recorded in the spreadsheet software Microsoft Office Excel (Microsoft, Redmond, WA, USA) (2010). Descriptive data are presented as mean ± standard deviation (SD).

### Surgical Procedure

The procedure is usually performed under general anesthesia in our department. After sterile draping, the affected canaliculus is first probed with a Bowman or Jünemann probe. The probe is used for orientation and splinting during the incision of the canaliculus. The canaliculus is then incised about 2 mm medial to the lacrimal punctum along its posterior wall, sparing the lacrimal punctum itself. Usually, pus or concretions will already be draining at this point. As bleeding occurs, a chalazion clamp can be applied. The remaining contents of the canaliculus are then curetted with a chalazion curette until no more concretions are recovered. The canaliculus is then rinsed extensively with iodine 5% solution. Subsequently, the entire nasolacrimal duct is rinsed with sodium chloride solution. Appendix A shows the surgical procedure up to this point.

Appendix A: Canaliculotomy with removal of pus and concretions from the superior canaliculus (Appendix A).

This is followed by bicanaliculonasal intubation with a silicone tube according to the atraumatic Münster intubation technique. After pulling the tube through to the ipsilateral nostril by using a thread-guide, the tube ends are knotted and shortened. The silicone tubes at our specialized center are usually kept in place for 3–6 months. This time interval represents the expert recommendation for nasolacrimal duct recanalization with tube intubation in patients with nasolacrimal duct obstruction in order to avoid postoperative adhesions [12,13]. However, to our knowledge, there are no recommendations on the duration of STI after canaliculotomy. Immediately after the procedure and for the following 3–5 days postoperatively, astringent (tetryzolin) and antibiotic eye drops (ofloxacin) are applied three times daily. The silicone tubes are usually kept in place for 3–6 months.

## 3. Results

A total of 25 eyes of 25 patients with canaliculitis who underwent canaliculotomy and STI could be included; 18 patients (72%) were female and 7 patients (28%) were male. The mean age was 63.7 ± 17.2 years. The mean follow-up period was 3.7 ± 1.5 years (44 ± 18 months) (Table 1). Overall, 22 of the 25 cases (88%) were free of recurrence at the time of follow-up.

The mean time to diagnosis was 19 ± 23.4 months. Time to diagnosis was defined as the period between the onset of the patient’s first symptoms and the diagnosis of canaliculitis. The data were taken from the patient’s history and/or the review of external findings provided. Figure 2 summarizes the average time to diagnosis for the study cohort. The mean duration of STI was 5.8 ± 3.4 months (174.7 ± 103.2 days). At the time of follow-up, 4 patients (16%) complained about epiphora in the treated eye at the time of follow-up.

Complications such as post-operative hemorrhage, spread of infection, obstruction of the canaliculus or migration of the STI were not observed in any of the included patients.

Tube dislocation occurred in 4 cases, resulting in recurrence in 2 patients. The 2 remaining dislocations proceeded without complications. Two of the patients with recurrence developed a pyogenic granuloma, and one of them also presented with a canalicular fistula. The first patient decided not to undergo another surgical procedure; the latter underwent surgical revision. Both patients were lost to follow-up. Another patient without recurrence, however, experienced refractory epiphora after external tube removal and needed a revision, which was due to incomplete removal of the tube material from the lacrimal sac. External dacryocystorhinostomy led to the complete resolution of symptoms.

Of the patients with recurrence (*n* = 3), one occurred after a duration of <1 year, and the others after a duration of >16 months. Two patients underwent a revision canaliculotomy with STI and have not experienced another recurrence to date. Table 2 displays detailed demographic and descriptive data on the patients who suffered a recurrence.

## 4. Discussion

Canaliculitis can severely affect a patient’s well-being, as it may cause distinct epiphora with continuous pus secretion, limiting everyday activities as well as participation in social and working life.

Surgical treatment should be the standard approach for the management of canaliculitis, as the success rates of surgical intervention are significantly higher at 80–100% than those of conservative treatment at about 60–70% [8,14,15,16]. The aim is a permanent removal of the concretions and sulfur granules from the canaliculus, which serve as a reservoir for pathogens. Canaliculotomy with curettage is often described in the literature as the procedure of choice because the surgical approach for the canaliculus, the canaliculotomy, allows the surgeon direct visualization and access to thoroughly remove the contents of the canaliculus [5,8,14,17]. However, the canaliculotomy itself can be associated with certain disadvantages and complications. Manipulation of the canaliculus may impair the function of the lacrimal apparatus postoperatively. In addition, fistula formation can result. Furthermore, consecutive scarring can lead to obstruction of the canaliculus and thus persistence of epiphora symptoms [1,8,14,15]. To prevent these complications, a nasolacrimal intubation with a silicone tube can be performed following a canaliculotomy with curettage. There is currently little data in the literature on this consecutive procedure [11]. Wang et al. recently compared the success and complication rates of canaliculotomy with and without STI for the first time and were able to demonstrate a significantly higher success rate in the group with STI (100% anatomical success) compared to the group without STI (78.3% anatomical success). At the same time, no postoperative canalicular obstruction occurred in the group with STI, compared to five cases (21.7%) in the group without STI. In this prospective study, the minimum follow-up was 1 year [11].

In the present study, 22 (88%) of the 25 patients with canaliculitis treated with canaliculotomy and STI showed no evidence of recurrence after a follow-up period of 3.7 ± 1.5 years. There was no secondary obstruction of the canaliculus in any case. However, in one case with recurrence, fistula formation was reported. Compared to Wang et al., the follow-up time in this study was significantly longer, with two late recurrences after 16 and 26 months, which would not have been included in the statistics of a follow-up of only 1 year.

To our knowledge, the cohort in this trial is the largest cohort of patients with canaliculitis evaluated for canaliculotomy with curettage and STI with a mean follow-up period greater than three years. A comparable long-term study by Lin et al., which describes canaliculotomy without STI, recorded a recurrence of canaliculitis in 7 (21%) of 34 patients after a 24-month follow-up period. Similarly, Anand et al. performed a long-term evaluation of incisional canaliculotomy with curettage without STI in 15 patients with canaliculitis and reported no recurrence in 100% of cases after a mean follow-up of 26 months. Two patients developed lacrimal duct obstruction postoperatively, without evidence of canalicular scarring. Both patients had a delayed diagnosis with multiple unsuccessful sac washouts [14]. Failure in these studies was defined as the recurrence of any inflammatory symptoms. We adapted this definition in our study. All in all, the data from this study and other trials suggest that the benefit-to-risk-ratio for the surgical treatment of canaliculitis can be regarded as very high.

Although canaliculotomy with STI shows very promising long-term results in our analysis and in the literature, more minimally invasive techniques for the treatment of canaliculitis should be mentioned and considered, such as access via the lacrimal punctum [18]. Lee et al. reported a reduction in symptoms after canaliculitis with one-snip punctoplasty followed by curettage in 25 (83%) of their 30 patients after an average follow-up period of about 11 weeks. Three of the five patients with recurrence then received conservative therapy with local antibiotics, and two underwent another curettage. Three weeks after the second procedure, they also showed no more symptoms of canaliculitis [18]. Pavilack et al. also advocate performing curettage via the lacrimal punctum without canaliculotomy. In a study of 11 patients with chronic canaliculitis, six patients received a one-snip punctoplasty, and five a singular curettage after dilatation of the lacrimal punctum. The procedure led to a reduction of symptoms in 100% of the patients. A revision procedure was required in five patients and a third procedure in one patient. The median follow-up time was three years [18].

In our patient population, revision surgery was required in a total of three cases: in two cases due to recurrence (one of which with fistula formation) and in one case due to an externally performed, incomplete tube removal. In the latter case, the remaining tube material had to be surgically removed from the lacrimal sac.

The limitations of the present study are its retrospective study design, which limits the comparison to other surgical procedures. Prospective, randomized studies that compare punctoplasties and canaliculotomies with STI, particularly in the long-term, would be desirable at this point.

Moreover, due to the absence of a standardized protocol in the follow-up-period, the differences in the recurrence may also be due to demographic, behavioral and clinical differences between patients. A prospective, randomized study design with a uniform definition of recurrence (e.g., anatomical versus functional) could also help to reliably compare surgical methods for the management of canaliculitis. Yet, measuring success by the subjective symptoms of the patient is a widely used tool in the evaluation of lacrimal duct surgery and the most important factor for patient satisfaction [19,20].

Based on the data at hand, it can be summarized that canaliculotomy with STI is a promising surgical approach with a low complication rate for the successful long-term treatment of canaliculitis. However, possible procedure-related complications such as fistula formation or improper handling of the STI should be kept in mind. Further long-term data with a uniform study design, especially for comparison with other more minimally invasive methods, are desirable.

## Figures and Tables

**Figure 1 jcm-11-06830-f001:**
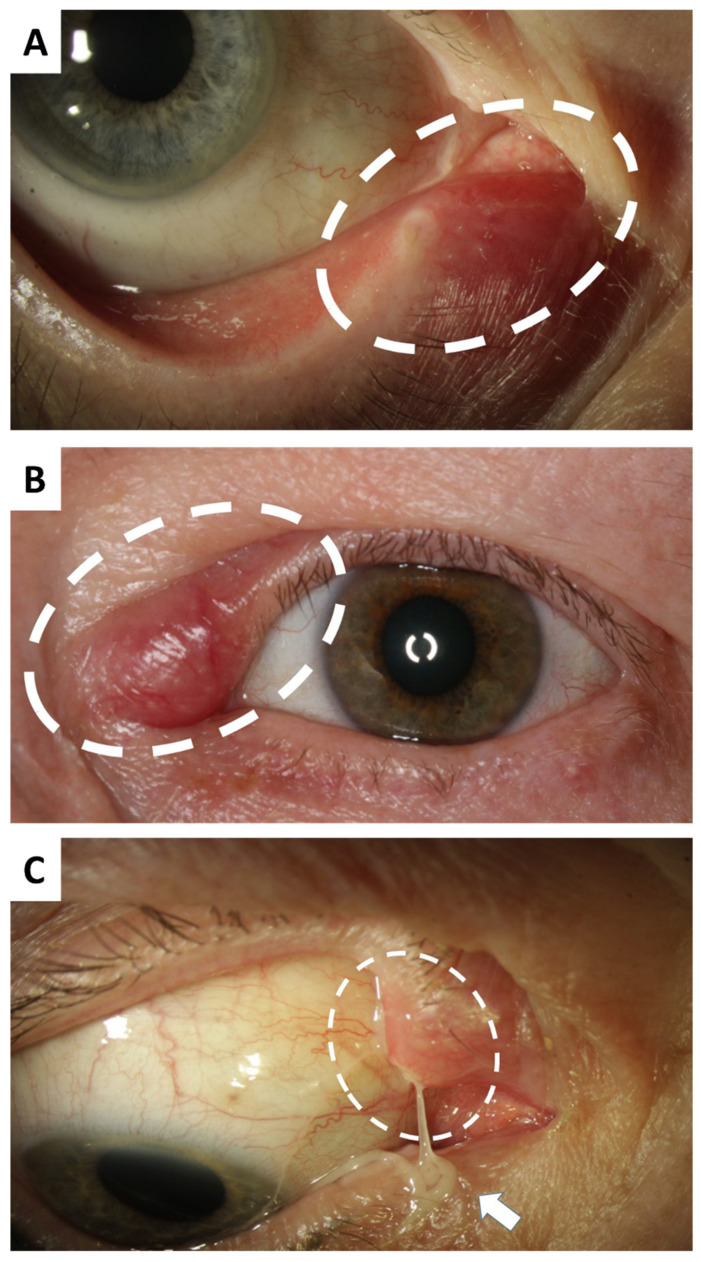
Clinical findings in canaliculitis. (**A**) Canaliculitis of the right inferior canaliculus in a 74-year-old female patient with complaints over the course of several weeks. Mild ectasia and redness are clinically visible (circled). (**B**) Left superior canaliculitis in a 66-year-old female patient with marked ectasia, swelling and redness (circled). The duration until the correct diagnosis was 4 months. (**C**) Canaliculitis of the right superior canaliculus of a 75-year-old female patient with pus discharge (arrow) from the upper ectatically altered lacrimal punctum (circled). The symptoms had been present for 12 months.

**Figure 2 jcm-11-06830-f002:**
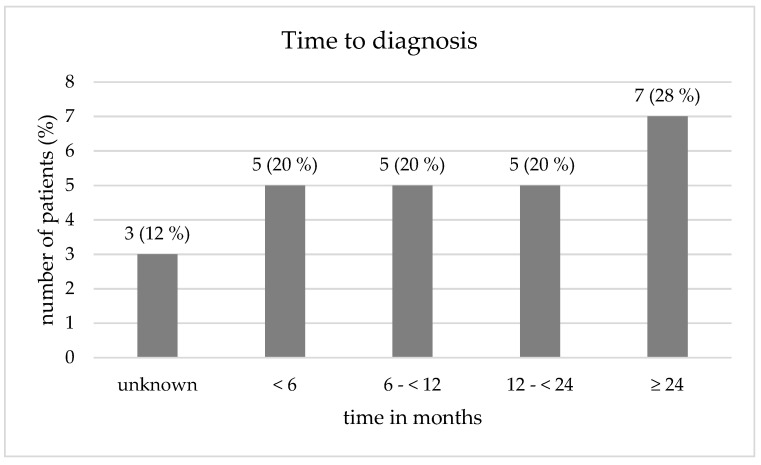
Time to diagnosis of canaliculitis in the study cohort (*n* = 25).

**Table 1 jcm-11-06830-t001:** Characteristics of study participants.

*n*	25
age (mean ± SD)	63.7 ± 17.2
gender (f/m)	18 (72%)/7 (28%)
location of canaliculitis (superior/inferior)	10 (40%)/15 (60%)
duration of follow-up in years (mean ± SD)	3.7 ± 1.5
duration of silicone tube intubation in months (mean ± SD)	5.8 ± 3.4
time from first symptoms to diagnosis in months (mean ± SD)	19 ± 23.4
number of patients without recurrence at follow-up	22/25 (88%)

*n* = number of patients; f = female; m = male; SD = standard deviation.

**Table 2 jcm-11-06830-t002:** Characteristics of patients with recurrence of canaliculitis (*n* = 3).

	Patient 1	Patient 2	Patient 3
age at time of recurrence (in years)	51	69	77
gender	female	female	female
affected canalicule	left inferior	right superior	right inferior
complications	STI dislocation, fistula formation, pyogenic granuloma	STI dislocation	pyogenic granuloma
time from first symptoms to diagnosis (in months)	6	12	4
duration of STI (in months)	1.2	5.9	8.5
time until recurrence (in months)	26	11	16
revision surgery (re-canaliculotomy)	Yes, recurrence-free for >56 months	Yes, recurrence-free for >42 months	Unknown

STI = silicone tube intubation.

## Data Availability

Not applicable.

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
