# Peer review of "Long-Term Outcomes of Canaliculotomy with Silicone Tube Intubation in the Management of Canaliculitis"

_jcm, 2022, doi:10.3390/jcm11226830_

Round 1

Reviewer 1 Report

The author investigated the rate of recurrence of chronic canaliculitis after canaliculotomy with silicon tube. Manuscript is well written and presents the data of this retrospective trial clearly. Data of 25 patients are presented. Comparison to a matched control group would be of great interest (e.g., canaliculotomy without STI).  

During review only a few minor comments arose

Minor Comments

There are several typos or formatting errors. E.g. L96, L102, L172, L208; please revise the manuscript carefully. 

Please comment and add some discussion about the decision when to remove the STI. How do you decide to remove it and when?

Tube dislocation means unintended removal?

How do you get the date for Time to diagnosis. Please comment. 

L164 you can leave out “(Table 2)”

L120 same for “(Video 1)

Reviewer 2 Report

General Comments

In this article, the authors carry out a study in which they analyze the efficacy of canaliculotomy in the treatment of canaliculitis.

This study is interesting due to the little information that there is on the therapeutic approach to this pathology.

Attached suggestion of changes that should be addressed to improve the manuscript.

Specify Comments

Although the size of the sample is small, it is understood by the little casuistry that there is of this pathology.

# 1 Introduction.

-       It is said that it is a rare disease, but apart from the percentage of lacrimal diseases, what percentage of the general population suffers from this pathology? Please explain.

-       Specify if it is monocular, binocular and why.

# 2 Materials and Methods.

-       Indicate if the patients had had any previous treatment.

-       Indicate the time of evolution of the pathology of the sample.

-       How were the complications referred to resolved? Please explain.

# 3 Discussion.

-       Clarify how limiting this pathology is for the patient.

-       Assess the benefit/risk ratio of surgery.
